# Real-World Analysis of Nivolumab and Atezolizumab Efficacy in Previously Treated Patients with Advanced Non-Small Cell Lung Cancer

**DOI:** 10.3390/ph15050533

**Published:** 2022-04-25

**Authors:** Miriam Alonso-García, Amparo Sánchez-Gastaldo, Miguel A. Muñoz-Fuentes, Sonia Molina-Pinelo, Laura Boyero, Johana Cristina Benedetti, Reyes Bernabé-Caro

**Affiliations:** 1Medical Oncology Department, Hospital Universitario Virgen del Rocío, 41013 Seville, Spain; miriamag3@hotmail.com (M.A.-G.); asanchezgastaldo@gmail.com (A.S.-G.); benedettipedroza@gmail.com (J.C.B.); 2Institute of Biomedicine of Seville (IBiS), HUVR, CSIC, Universidad de Sevilla, 41013 Seville, Spain; miguelmf8145@gmail.com (M.A.M.-F.); pinelo_sonia@hotmail.com (S.M.-P.); lboyero@hotmail.com (L.B.); 3CIBERONC, 28029 Madrid, Spain

**Keywords:** atezolizumab, immune checkpoint inhibitors (ICIs), immunotherapy, nivolumab, non-small cell lung cancer (NSCLC), real-world data

## Abstract

Nivolumab (anti-PD-1 antibody) and atezolizumab (anti-PD-L1 antibody) have shown superior survival outcomes and improved adverse effects compared to standard chemotherapy in advanced non-small cell lung cancer (NSCLC) patients. However, the efficacy of both treatments has not been directly compared in clinical trials. This retrospective, single-centre study was performed from June 2015 to December 2020 and included a cohort of 158 previously treated patients with stage IV or recurrent NSCLC who received PD-1 (nivolumab) (*n* = 89) or PD-L1 (atezolizumab) (*n* = 69) inhibitors at the Virgen del Rocío Hospital in Seville. The objective response rate (ORR) was 22.5% in the nivolumab group and 14.5% in the atezolizumab group (*p* = 0.140). Multivariate analysis did not show significant differences between the two groups for PFS and OS (PFS hazard ratio (HR): 0.80, 95% confidence interval (CI): 0.55–1.17, *p* = 0.260; OS HR: 0.79, 95% CI: 0.52–1.21, *p* = 0.281). Adverse events of all grades occurred in 68 patients in the nivolumab group (76.4%) and in 34 patients in the atezolizumab group (49.3%) (*p* < 0.001). Atezolizumab and nivolumab did not show statistically significant differences in survival outcomes in patients with NSCLC, even when stratified by histological subtype (squamous versus nonsquamous). However, the safety analysis suggested a more favourable toxicity profile for atezolizumab.

## 1. Introduction

Lung cancer remains the most common cause of cancer-related death worldwide [1]. Among all types of lung cancer, non-small cell lung cancer (NSCLC) accounts for approximately 85% of all cases. In recent years, immunotherapy agents, specifically immune checkpoint inhibitors (ICIs) that target programmed death 1 (PD-1) or its ligand (PD-L1), have shown revolutionary benefits against NSCLC, among other solid tumours [2]. Between 2015 and 2016, three agents, nivolumab, pembrolizumab (anti-PD-1 antibodies) and atezolizumab (anti PD-L1 antibody), were approved for the treatment of advanced NSCLC and are now widely used in clinical practice. These three ICIs were compared to standard chemotherapy, docetaxel, in a second-line treatment for advanced NSCLC in randomized phase 3 clinical trials [3,4,5,6]. In all of these studies, a significant improvement in overall survival (OS), objective response rate (ORR) and duration of response to ICIs over docetaxel was observed, although there was no clear difference when progression-free survival (median PFS) was compared. Equally important is the improvement in the quality of life of patients treated with ICI as these treatments present a better safety profile compared to chemotherapy.

Among the most common histological subtypes in NSCLC, non-squamous accounts for approximately 70% of all NSCLC, and squamous cell lung cancer accounts for approximately 30%. The prognosis of patients with squamous histology is worse than for those with adenocarcinoma histology [7]. The results obtained in the randomized phase 3 clinical trials (Checkmate 017, Checkmate 057 and OAK) indicated that ICIs significantly improved OS in patients with squamous and nonsquamous NSCLC; however, the efficacy was different between the two histological subtypes. In this matter, a subgroup analysis suggested that the antitumour efficacy of nivolumab versus docetaxel was higher for squamous NSCLC than for nonsquamous NSCLC in terms of PFS and OS. Thus, nivolumab showed a 41% lower risk of death (HR: 0.59, 95% CI: 0.62–0.87) in squamous patients (Checkmate-017), while in nonsquamous patients (Checkmate-057), it was 27% (HR: 0.73, 95% CI: 0.59–0.89). However, the possible differences in ICI treatments between the two main subtypes have not yet been fully evaluated in the real world, and there is only indirect information obtained from clinical trials.

Following approval of the treatments by the competent authorities (such as Food and Drug Administration (FDA) and European Medicines Agency (EMA), among others) for use in advanced NSCLC, numerous retrospective observational studies have been published evaluating the efficacy and safety of second-line or subsequent immunotherapy in the real world (Table 1) [8]. In many of these studies, inferior survival and safety outcomes have been obtained compared with pivotal studies, due in part to strict inclusion and exclusion criteria for patients in trials such as Eastern Cooperative Oncology Group performance status (ECOG-PS) scores below 2, patients without central nervous system (CNS) metastases or autoimmune disease. In addition, the median age of the patients in the clinical trials is usually lower than that found in clinical practice. All these factors hinder the studies’ capacity to reflect the entire cancer population. However, the results obtained in real-world studies tend to confirm an improvement in both efficacy and safety with respect to chemotherapy. In the current literature, the most widely studied ICI is nivolumab, and in this way, Barlesi et al. in a prospective study with 1420 NSCLC patients confirmed both the effectiveness and safety of nivolumab observed in clinical trials for the treatment of advanced NSCLC in a real-life setting in France, even including a high percentage of patients with ECOG ≥ 2 (17.1%) and with CNS metastasis (19.9%) [9]. Regarding atezolizumab, Furuya et al. in a multicentre study with 152 NSCLC patients demonstrated good efficacy and safety regardless of heavily treated patients, and ECOG = 0 was a favourable predictive factor [10]. However, few publications have investigated the efficacy of atezolizumab monotherapy in clinical practice or compared the outcomes of nivolumab (anti-PD-1 antibody) and atezolizumab (anti-PD-L1 antibody). To our knowledge, only two studies compared them. Ramagopalan et al. [11] studied them in a large cohort that compared the effectiveness (OS), and Weis et al. [12] compared survival, response and toxicity, concluding in both cases that there was no statistically significant difference between the two treatments. With this aim, the main objective of this work is to characterize and compare the results (survival, response and safety) between nivolumab and atezolizumab in pretreated (second- and third-line treatment) patients with advanced NSCLC. Likewise, the effectiveness of both drugs is studied when patients are stratified by histological subtype (squamous and nonsquamous).

## 2. Results

### 2.1. Patient Characteristics

A total of 158 patients with advanced NSCLC who were treated with nivolumab (*n* = 89) or atezolizumab (*n* = 69) as a second- or third-line treatment were enrolled in the study. The clinicopathological characteristics of all the patients are described in Table 2. The median age at the beginning of ICI treatment was 64 years (range 37–86). It was significantly higher in patients treated with nivolumab (*p* = 0.005). The majority of patients were male (78.5%), and only 15.2% had an ECOG-PS greater than or equal to two. For the whole cohort, a similar distribution pattern represented by histological subtype was represented. However, this distribution varied significantly according to immunotherapy type, and there were more patients with squamous histology in the nivolumab group (*p* < 0.001). Baseline CNS metastases were found in 16.5% of the patients, with a significant difference between the nivolumab and atezolizumab groups (*p* = 0.044). There were more patients with analysed PD-L1 expression in the atezolizumab than in the nivolumab group (*p* < 0.001). Most patients were current smokers or had quit smoking less than 10 years ago (73.4%). There were more patients who received ICI as the second-line of treatment (79.1%) and who were previously treated with platinum-based therapies (94.9%). The median length of follow-up for the entire cohort was 8.32 months (95% CI: 0.10–46.95). On the other hand, Appendix A shows the baseline characteristics of the patients grouped by histological type; in this case, the age was significantly higher in the squamous group (*p* < 0.001), as well as the presence of COPD (*p* = 0.041), while the presence of CNS metastasis was significantly higher in the nonsquamous group (*p* = 0.004).

By the end of the study, 110 patients (69.6%) died, 64 (71.9%) were included in the nivolumab group and 46 (66.7%) were included in the atezolizumab group. After ICI discontinuation (Table 3), 62 (39.24%) patients received conventional chemotherapy, and one patient received TKI (erlotinib) therapy. There was no difference in the administration of the subsequent treatment between the two groups. Vinorelbine and docetaxel were the most commonly used subsequent treatments, 14.6% and 13.9%, respectively.

### 2.2. Efficacy Outcomes

To evaluate the efficacy outcomes of nivolumab and atezolizumab in pretreated patients with advanced NSCLC, we analysed their response according to the Response Evaluation Criteria in Solid Tumours version 1.1. The objective response rate (ORR) and disease control rate (DCR) for the whole cohort were 19% and 43.7%, respectively (Table 4). Complete response (CR) was achieved in three patients in the nivolumab group. There were no statistically significant differences between the nivolumab and atezolizumab groups with respect to ORR (22.5% vs. 14.5%, *p* = 0.140) and DCR (49.4% vs. 36.2%, *p* = 0.113). Response could not be assessed in one patient in the atezolizumab group.

### 2.3. Survival Outcomes

For the whole cohort (Figure 1), the median PFS and OS values were 3.19 months (95% CI: 2.31–4.06) and 9.03 months (95% CI: 6.60–11.47), respectively. The estimated rate of PFS was 37.1% at 6 months and 27.1% at 12 months. For OS, the estimated rate was 41.9% at 12 months, 27.5% at 24 months and 15.9% at 36 months. The median PFS for patients treated with nivolumab was 3.55 months (95% CI: 1.66–5.44), while the median PFS for those treated with atezolizumab was 2.89 months (95% CI: 1.84–3.94) (Figure 2A). The median OS in the patients treated with nivolumab and atezolizumab was 9.03 (95% CI: 5.51–12.56) and 9.00 (95% CI: 6.00–12.00), respectively (Figure 2B). There were no significant differences in either PFS (*p* = 0.496) or OS (*p* = 0.685) between patients treated with nivolumab and those treated with atezolizumab.

The univariate analysis (Table 5) showed that an ECOG of 0–1 (HR: 0.47, 95% CI: 0.29–0.77, *p* = 0.003) was associated with a longer PFS, and nonsquamous histology (HR: 1.48, 95% CI: 1.02–2.16, *p* = 0.040) and an ECOG of 0-1 (HR: 0.38, 95% CI: 0.22–0.64, *p* < 0.001) were associated with a longer OS. We did not find evidence of an association between PFS or OS and age, sex, PD-L1 expression, CNS metastasis, smoking status, treatment, line of therapy or initial platinum therapy. Variables with a *p* value <0.05 in the univariate analysis as well as treatment (nivolumab vs. atezolizumab) were used in the multivariate analysis (Table 6). In this case, as in the univariate analysis, only an ECOG of 0-1 (HR: 0.49, 95% CI: 0.30–0.79, *p* = 0.004) for PFS and a nonsquamous histology (HR: 1.60, 95% CI: 1.12–2.53, *p* = 0.012) and ECOG of 0-1 (HR: 0.37, 95% CI: 0.22–0.63, *p* < 0.001) for OS were associated with better outcomes.

Finally, as in our study, age, histology, CNS metastasis and PD-L1 status were significantly different between the two ICIs, and we performed an analysis according to these variables (Appendix A). Our results showed that there were no differences in PFS or OS across the different subgroups between the atezolizumab and nivolumab cohorts.

#### Survival Analysis According to Histological Subtype (Squamous versus Nonsquamous)

Figure 3 shows the Kaplan–Meier curves according to the histological subtype. Squamous (*n* = 71) and nonsquamous (*n* = 87) histology showed no statistically significant differences with respect to PFS and OS between nivolumab (squamous (*n* = 56), PFS: 3.25 months; 95% CI: 1.25–5.18 and OS: 7.88 months; 95% CI: 3.67–12.09; nonsquamous (*n* = 33), PFS: 3.91 months; 95% CI: 0.00–9.52, OS: 9.85; 95% CI: 4.16–15.55) and atezolizumab (squamous (*n* = 15), PFS: 3.45 months; 95% CI: 1.71–5.92 and OS: 5.58 months; 95% CI: 3.47–7.70; nonsquamous (*n* = 54), PFS: 2.76 months; 95% CI: 1.76–3.75 and OS: 9.72 months; 95% CI: 5.01–14.44).

### 2.4. Safety Outcomes

The total number of patients with at least one adverse event (AE) was 102 (64.55%) (Table 7). When stratified by treatment, the AE analysis suggested a less favourable toxicity profile of nivolumab compared to atezolizumab (76.4% vs. 49.3%; *p* < 0.001). Severe AEs, grade 3 or 4, only occurred in 14.55% of patients (19.10% and 8.69% in the nivolumab and atezolizumab groups, respectively). No AEs resulting in death were reported; however, in the whole cohort, 9.49% of patients, 11.23% in the nivolumab group and 7.24% in the atezolizumab group, suffered an AE that led to the definitive discontinuation of treatment. The most common were asthenia/fatigue, 53.93% and 31.88%, and anorexia/weight loss, 17.98% vs. 7.24% in the nivolumab and atezolizumab groups, respectively. Regarding immune-related adverse effects, the most common were skin disorders (including pruritus and RASH), 33.70% vs. 18.84%, and pneumonitis, which were experienced by 9% and 2.9% in the nivolumab group and the atezolizumab group, respectively.

## 3. Discussion

Atezolizumab and nivolumab, PD-L1 and PD-1 antibodies, respectively, are proven to be effective and safe in clinical trials [3,4,6] and after approval by authorities in real-world studies (Table 1) [9,10,12,13,14,15,16,17]. However, there is no head-to-head clinical trial that compares the results of each drug, and to our knowledge, there are only two real-world studies that have compared outcomes between the two ICIs [11,12]. However, they did not perform a head-to-head subgroup analysis between the two ICIs. In the present study, we evaluated and compared the efficacy and safety of two immune checkpoint inhibitors, atezolizumab and nivolumab, as second- or third-line therapies in the control of advanced NSCLC in real-world clinical practice. Likewise, a subgroup study was conducted comparing the efficacy in terms of PFS and OS of both ICIs.

The results of our analysis are slightly lower than those reported in clinical trials for mOS but not for mPFS. The median PFS and OS reported for ICIs in real-world studies are heterogeneous, ranging from 1.9 to 6.1 months and 6.5 to 18 months, respectively. One of the strengths of immunotherapy has been the increase in OS, thus increasing patients’ long-term survival. In this respect, numerous studies have shown an increase in the effectiveness of post-ICI chemotherapy [18,19]. Regarding this, the estimated rate of OS in our study was 15.9% at 36 months. The 3-year OS rates in a combined analysis for CheckMate 017 and 057 studies [20] were 17.1% for patients receiving nivolumab and 21% for patients receiving atezolizumab [21]. Therefore, in this case, our results are also slightly lower than those of clinical trials. Meanwhile, the objective response rate (ORR) in our study was 19%, similar to that reported in the CheckMate-017 (20%) or CheckMate-057 (19%) and higher than that obtained in the OAK study (14%). In real-world data, the ORR ranged from 8.6% to 28.57%. This contrast in the real-world outcomes (PFS, OS and ORR) is due to several factors, such as the inclusion of a very heterogeneous population between studies that include patients who are generally excluded from clinical trials, including elderly patients, heavily pretreated, patients with an ECOG score greater than or equal to 2, CNS metastasis and/or with numerous comorbidities. For example, poor survival outcomes have been obtained in studies that include a high number of patients with negative prognostic factors such as ECOG ≥ 2 or metastases in the CNS [12,22,23,24]. In our cohort, 15.2% of patients had ECOG ≥ 2, and 16.5% had CNS metastases at baseline, similar to other real-world studies [9,16,17] and higher than those included in the pivotal studies, 1.5%, 0% and 0% for ECOG ≥ 2 and 6.6%, 11.6% and 10% for CNS metastases in Checkmate017, Checkmate057 and OAK, respectively. However, in our study, only ECOG was an independent prognostic factor for survival outcomes (PFS and OS). Our results are in agreement with several analyses showing that ECOG-0-1 is the most significant predictor of clinical benefit [14,25]. Another important factor influencing the contrast found in response in real-world studies is the possible bias when evaluating the response to treatment as, in many cases, clinicians rather than independent radiological reviewers assess it.

Although there are no head-to-head comparisons between nivolumab and atezolizumab, some meta-analyses and systematic reviews have been performed to evaluate the efficacy and safety between the different ICIs. Passiglia et al. [26] performed a meta-analysis of all phase II/III randomized clinical trials comparing PD1/PDL1 inhibitors versus docetaxel in pretreated NSCLC patients. The study conducted an indirect comparison between the differences in efficacy and safety profiles between atezolizumab, pembrolizumab and nivolumab, concluding that nivolumab and pembrolizumab were associated with a significant increase in ORR compared to atezolizumab (Nivo vs. Atezo HR 1.66, 95% CI 1.07–2.58), but no statistically significant differences were found in PFS or OS. Regarding safety, nivolumab was associated with a significantly lower risk for G3/G5 AEs. On the other hand, Liang et al. [27] conducted a meta-analysis that included 19 randomized clinical trials of anti-PD-1/L1 according to the treatment line with 11456 patients with advanced NSCLC. From that analysis, they concluded that nivolumab was the best option among PD-1/PD-L1 inhibitors for patients in second or further lines (Nivo vs. Atezo, PFS HR = 0.84; 95% CI (0.71–0.99); ORR RR = 1.73; 95% CI (1.16–2.58)), while atezolizumab is the most tolerable in terms of severe AEs; therefore, it is an alternative for patients with poor clinical conditions. Concerning the real world, Mencoboni et al. [8] performed a meta-analysis that enrolled 32 studies, most of which were treated with nivolumab, concluding that the efficacy and safety of ICIs in clinical practice are comparable to those in clinical trials. Recently, Ramagopalan et al. [11], in a real-world study of a large cohort (3336 patients), did not find significant differences in OS between atezolizumab and nivolumab in the overall population or in any subgroup that was examined. Similar results were reported by Weis et al. [12] in a smaller cohort. In our study, when we compared atezolizumab to nivolumab, the survival analysis showed that there were no differences in terms of efficacy (response, PFS and OS) between the two treatments.

When we analysed the baseline variables that were different between both treatments, we did not obtain significant differences neither in the PFS nor in the OS. Therefore, the clinical outcome is independent of the baseline characteristics of each study cohort prior to initiation of immunotherapy. Furthermore, the univariate analysis did not show significant differences between these subgroups. Regarding age, most of the studies that analysed it have reported that elderly patients with a good ECOG score have the same benefits as young people in terms of efficacy and safety [28,29,30]. In our study, most of the elderly patients had an ECOG 0-1 (88.2%) at the beginning of immunotherapy, and there was no significant difference in the distribution of this subgroup with respect to ECOG between both treatments. We also found no significant effect of PD-L1 expression, although the analysis may have been underpowered to detect an association given that the sample size of patients for whom PD-L1 expression data were not available was large (38.6%). Finally, the multivariate analysis did not modify the interpretation of the results obtained in the univariate analysis. In our study, only the ECOG score for PFS and OS and histology for OS were significant variables, concluding that no significant differences were found between the two ICIs with respect to survival outcomes.

To date, there are no real-world studies comparing ICIs stratified by histology. Most of the studies that have analysed histology have been with nivolumab and have found no differences in efficacy between squamous and nonsquamous tumours [14,29,31]. In our study, most patients with a squamous histology were treated with nivolumab, which is based on the fact that in pivotal studies, nivolumab showed a 41% lower risk of death (HR: 0.59, 95% CI: 0.62–0.87), while atezolizumab showed 27% (HR: 0.73, 95% CI: 0.54–0.98), when both were compared with docetaxel [3,6]. However, we did not find differences in terms of the efficacy of both drugs when we stratified NSCLC patients by histological subtype.

Concerning safety, in our cohort, patients who experienced AEs that led to permanent drug discontinuation were slightly higher than those found in CheckMate-017, CheckMate-057 and OAK, where the permanent discontinuation rates were 3%, 5% and 8%, respectively. The inclusion in our study of patients with ECOG ≥ 2, elderly and with numerous comorbidities and metastasis, as well as a possible inadequate management of AEs, could explain this higher discontinuation rate. In addition, the AE analysis suggested a less favourable toxicity profile of nivolumab compared to atezolizumab. In this sense, Liang et al. [27] reported that atezolizumab is better at reducing severe AEs in patients without severe pulmonary symptoms than nivolumab. However, Duan et al. [32], in a meta-analysis of 19 clinical trials, found no significant differences in toxicity between anti-PD-1 and anti-PD-L1 treatments. Likewise, Weis et al. [12] also found no differences between atezolizumab and nivolumab. The most common AEs reported were anorexia/weight loss, asthenia/fatigue and skin disorder (including pruritus and rash). The safety profile observed herein is consistent with data from pivotal clinical trials and other real-world studies. Finally, in relation to the higher toxicity among patients receiving nivolumab, it should be noted that three times more patients with squamous cell carcinoma were treated with nivolumab than with atezolizumab. Patients with squamous cell carcinoma are usually heavy smokers and often have been associated with significant pulmonary and vascular comorbidity [7]. In our study, the squamous group was older and had a higher percentage of patients with COPD. This factor, together with the fact that the majority of patients treated with nivolumab were older compared to atezolizumab, could explain the difference in toxicity.

Ultimately, the clinical importance of this study lies in the lack of head-to-head clinical trials of anti-PD-1 and anti-PD-L1 that allow us to compare the efficacy and safety of both drugs, thus deciding the best therapeutic option for NSCLC patients. However, this was a retrospective, nonrandomized, single-centre study, and it had several limitations that must be taken into account when interpreting the results. First, the patient sample size was small and bias could have existed in efficacy and safety evaluations. Second, the retrospective and observational nature of the study means that patients were not assigned to treatment groups and underlying clinical differences may have influenced the results. In this way, the sample size was also unbalanced between the treatment groups, with a substantially higher proportion of patients receiving nivolumab. Third, electronic medical records, the data source, were designed for documentation and management of clinical practice, not for research, so information such as low-grade adverse effects could have been underestimated or overlooked. Fourth, some patients were not evaluated with regular computed tomography, due to clinical practice, and the frequency of disease evaluation in each patient could be different, which means that PFS could not be strictly evaluated. Fifth, comorbidities and metastasis have not been taken into account in survival studies. Sixth, 38.6% of the patients in our study had an unknown expression of PD-L1, a biomarker that has consistently indicated a response to immunotherapy in clinical trials, which may have affected the efficacy results of nivolumab and atezolizumab.

## 4. Materials and Methods

### 4.1. Patients and Study Design

This retrospective study performed from June 2015 to December 2020 aimed to characterize and compare the results between nivolumab and atezolizumab in pretreated advanced NSCLC patients (second- and third-line treatment) at the Virgen del Rocio Hospital in Seville. The inclusion criteria were (i) confirmed stage IV or recurrent NSCLC who progressed during or following first-line treatment and (ii) use of nivolumab or atezolizumab as second- or third-line treatment. The exclusion criteria were (i) previous ICI (alone or in combination with chemotherapy) therapy, (ii) ICI given within clinical trials, and (iii) patients treated with ICI after treatment with nivolumab or atezolizumab (Figure 4). Only patients who could achieve a potential minimum follow-up of 6 months were selected for the study. Thus, this study included a cohort of 158 subjects treated with nivolumab (*n* = 89) or atezolizumab (*n* = 69). Patients receiving nivolumab were administered doses of 240 mg or 3 mg/kg every two weeks or 480 mg every four weeks. In the atezolizumab group, patients received a dose of 1200 mg every three weeks. The clinicopathological and demographic characteristics of the patients were collected through 31 December 2020. Treatment selection (atezolizumab or nivolumab) was based on its availability and the frequency of administration. In many cases, the preference for receiving 3-week cycles of atezolizumab versus 4-week cycles of nivolumab has played a role in treatment decisions. Finally, nivolumab has been available in Spain for second-line treatment since 2015, while atezolizumab was incorporated in 2018 in this indication. For the evaluation of adverse events (AEs) in our centre, clinicians classify them according to the NCI Common Terminology Criteria for Adverse Events (CTCAE) version 4.0. In addition, tumour response was evaluated by computed tomography and assessed with Response Evaluation Criteria in Solid Tumours version 1.1 [33] by radiologists and medical oncologists. The objective tumour response rate (ORR) was considered the best clinical response during the course of treatment, including complete response (CR) and partial response (PR), disease control rate (DCR) defined by patients displaying stable disease (SD), partial response (PR) or complete response (CR), and progressive disease (PD). PD-L1 expression was evaluated by immunohistochemical staining using PD-L1 monoclonal antibodies (VENTANA PD-L1 (SP263) assay by Roche).

### 4.2. Statistical Analysis

Descriptive analyses were used to characterize the most relevant clinical variables. Categorical parameters were explored using the chi-squared test or Fisher’s exact test. Age (≤65 vs. >65), ECOG score (0–1 vs. ≥2) and smoking status (never or +10 years former smokers vs. former or −10 years former smokers) were modelled as categorical (binary) variables. Continuous variables were compared using Student’s t test or the Mann–Whitney U test depending on whether they showed a normal distribution.

For survival studies, PFS was measured from the time of initiating nivolumab or atezolizumab treatment to clinical or radiographic progression or death from any cause or was censored on the day of cut-off. OS was measured from the time of initiating nivolumab or atezolizumab treatment to death from any cause or was censored on the day of cut-off. OS and PFS were calculated using the Kaplan–Meier method and log-rank test. Hazard ratios (HRs) and 95% confidence intervals (CIs) were calculated using the univariate Cox proportional hazard model. Parameters with a value of *p <* 0.05 (considered statistically significant) in the univariate analysis were selected for inclusion in the multivariate analysis. The median follow-up time was calculated using all patients and was estimated from observed follow-up times. Statistical analyses were performed using Statistical Package for the Social Sciences software (SPSS version 20, Chicago, IL, USA).

## 5. Conclusions

In this real-world study, we found that the safety and efficacy of nivolumab and atezolizumab in previously treated patients with advanced NSCLC are consistent with those found in randomized clinical trials and comparable to the results from other real-world studies. In addition, we identified that atezolizumab and nivolumab showed similar survival outcomes in NSCLC patients, even when they were stratified by histological subtypes (squamous versus nonsquamous). However, safety analysis suggested a more favourable toxicity profile for atezolizumab.

## Figures and Tables

**Figure 1 pharmaceuticals-15-00533-f001:**
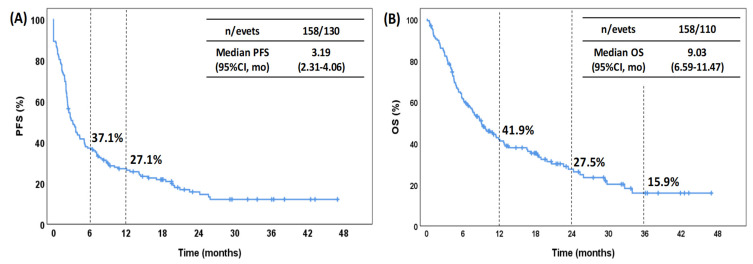
Kaplan–Meier curves for PFS (**A**) and OS (**B**) for the entire cohort. PFS, progression-free survival; OS, overall survival; CI, confidence interval; mo, months.

**Figure 2 pharmaceuticals-15-00533-f002:**
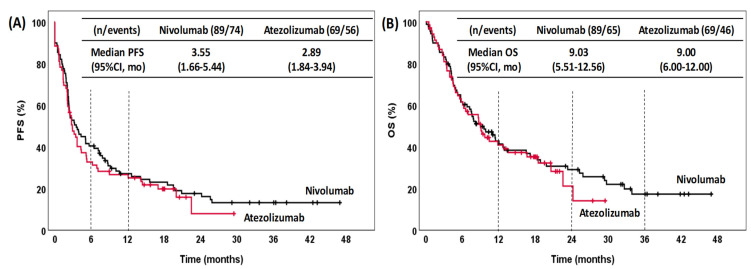
Kaplan–Meier curves for PFS (**A**) and OS (**B**) stratified by treatment. PFS, progression-free survival; OS, overall survival; CI, confidence interval; mo, months.

**Figure 3 pharmaceuticals-15-00533-f003:**
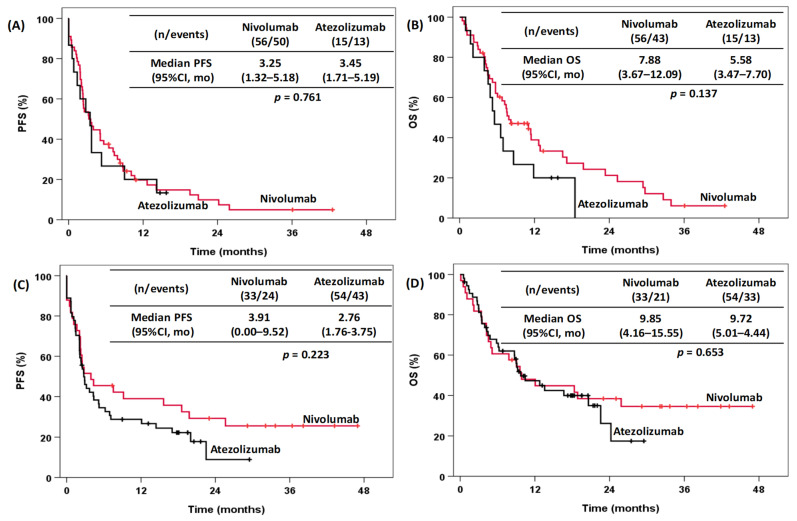
Kaplan–Meier curves for NSCLC patients treated with nivolumab and atezolizumab stratified according to squamous (**A**,**B**) and nonsquamous (**C**,**D**) histology. Left plots: (**A**,**C**) PFS curves, and right plots: (**B**,**D**) OS curves. PFS, progression-free survival; OS, overall survival; CI, confidence interval; mo, months.

**Figure 4 pharmaceuticals-15-00533-f004:**
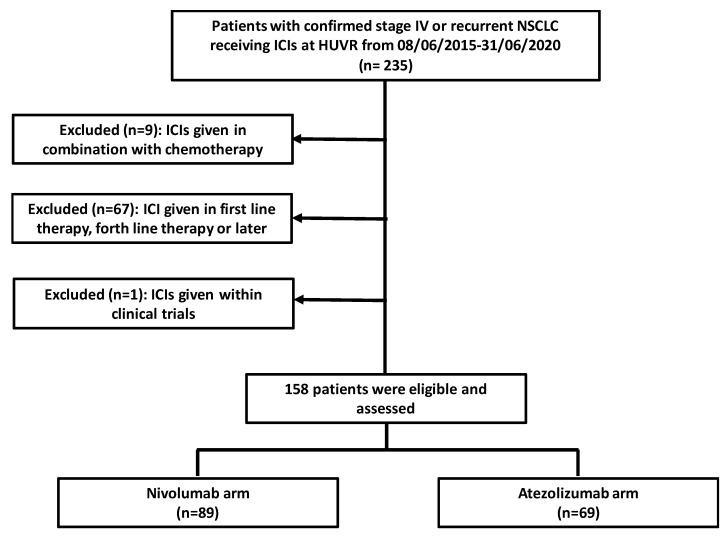
Flowchart of the patients selected for the study.

**Table 1 pharmaceuticals-15-00533-t001:** Characteristics and outcomes of the pivotal randomized clinical trials, real-world studies and present study.

Study	Treatment	Line	Population %	Patients *n*	Age, Median (Range)	ECOG ≥ 2 *n* (%)	CNS Met *n* (%)	PFS (Months)	OS (Months)	ORR (%)
Checkmate 017 [3]	Nivo	≥2	Squamous	135	62 (39–85)	2 (1.5)	9 (6.6)	3.5	9.2	20
Checkmate 057 [4]	Nivo	≥2	Non-sq	292	61 (37–84)	0	34 (11.6)	2.3	12.2	19
OAK [6]	Atezo	≥2	Sq: 26 Non-sq: 74	425	63 (33–82)	0	85 (10.0) ^a^	2.8	13.8	14
Ramagopalan et al., (2021) (USA) [11]	Nivo Atezo	1.31 1.35 ^b^	Sq: 30.82 Non-sq: 65.55 Other: 3.63	2630 206	67.28 (mean) 68.30 (mean)	739 (28.1) 50 (24.3)	191 (7.3) 37 (18)	nr	HR: 1.04 (95% CI: 0.88–1.24)	nr
Weis et al., (2020) (USA) [12]	Atezo Nivo	≥2	Sq: 30.65 Non-sq: 62.90 Other: 6.45	43 81	67.2 (nr) 64.3 (nr)	9 (20.9) 21 (28.4)	nr	2.0 2.2	6.5 8.4	13.9 14.8
Ivanovic et al., (2021) (Slovenia) [13]	Nivo, Atezo, Pembro.	2	Sq: 15 Non-sq: 85	40	63 (42–77)	1 (3.0)	8 (20.0)	3.5	9.9	25
Figueiredo et al., (2020) (Portugal) [14]	Nivo	≥2	Sq: 12 Non-sq: 88	219	64 (37–83)	29 (13.2)	nr	4.9	13.2	22.4
Chen et al., (2020) (China) [15]	Nivo Pembro.	≥2	Sq: 40 Non-sq: 60	62 35	64 (IQR:57–69)	nr (15)	nr	5	18	16.5
Martin et al., (2020) (Argentina) [16]	Nivo	≥1	Sq: 20.2 Non-sq: 78 Other: 1.8	109	65 (56–72)	17 (15.6)	nr	6.1	12.1	28.57
Furuya et al., (2021) (Japan) ^c^ [10]	Atezo	≥1	Sq: 13.16 ADC: 80.26 Other: 6.58	152	71 (43–93)	8 (5.26)	39 (25.65)	1.4	12.8	8.6
Barlesi et al., (2020) (France) [9]	Nivo	≥1 ^d^	Sq: 30.8 Non-sq: 69.2	1420	66 (35–91)	241 (17.1)	282 (19.9)	2.8	11.2	19.6
El Karak et al., (2019) (Lebanon) [17]	Nivo Pembro	≥2	Sq: 30 ADC: 57 Other: 13	55 55	66 (nr)	nr	17 (15.5)	4	8.1	25
Current study	Whole Cohort	2–3	Sq: 44.9 Non-sq: 55.1	158	64 (37–86)	24 (15.2)	26 (16.5)	3.19	9.03	19
	Nivo			89	66 (37–86)	14 (15.7)	10 (11.2)	3.55	9.03	22.50
	Atezo			69	62 (40–80)	10 (15.5)	16 (23.2)	2.89	9.00	14.49

^a^ Reported for atezolizumab and docetaxel cohort combined; ^b^ Mean of previous lines; ^c^ Includes patients pretreated with ICI; ^d^ Four (0.3%) patients in first line. ECOG: Eastern Cooperative Oncology Group; CNS met: Central Nervous System metastases; PFS: Progression-Free Survival; OS: Overall Survival; ORR: Objective Response Rate; Sq: Squamous; Non-sq: Nonsquamous; nr: not reported.

**Table 2 pharmaceuticals-15-00533-t002:** Baseline characteristics by treatment group.

Characteristic *n* (%)	All Patients (*n* = 158)	Nivolumab (*n* = 89)	Atezolizumab (*n* = 69)	*p* Value
Age. median (range). years	64 (37–86)	66 (37–86)	62 (40–80)	0.005 ‡
Binary age at ICI initiation				0.013 *
≤65	90 (57)	43 (48.3)	47 (68.1)	
>65	68 (43)	46 (51.7)	22 (31.9)	
Sex				0.401 *
Male	124 (78.5)	72 (80.9)	52 (75.4)	
Female	34 (21.5)	17 (19.1)	17 (24.6)	
ECOG score at ICI initiation				0.830 *
0–1	134 (84.8)	75 (84.3)	59 (85.5)	
≥2	24 (15.2)	14 (15.7)	10 (15.5)	
Histology				<0.001 *
Squamous	71 (44.9)	56 (62.9)	15 (21.7)	
Non-squamous	87 (55.1)	33 (37.1)	54 (78.3)	
CNS metastasis	26 (16.5)	10 (11.2)	16 (23.2)	0.044 *
COPD	45 (28.8)	26 (29.2)	19 (27.5)	0.817 *
PD-L1 (positive ≥ 1% tumour cells)				<0.001 *
Yes	60 (38.0)	27 (30.3)	33 (47.8)	
No	37 (23.4)	13 (14.6)	24 (34.8)	
Unknown	61 (38.6)	49 (55.1)	12 (17.4)	
Smoking status				0.184 *
Never or +10 years former smokers	42 (26.6)	20 (22.5)	22 (31.9)	
Current or −10 years former smokers	116 (73.4)	69 (77.5)	47 (68.1)	
Treatment lines				0.157 *
2	125 (79.1)	74 (83.1)	51 (73.9)	
3	33 (20.9)	15 (16.9)	18 (26.1)	
Initial Platinum therapy				1 †
Yes	150 (94.9)	84 (94.4)	66 (95.7)	
No	8 (5.1)	5 (5.6)	3 (4.3)	
Follow-up (95% CI, mo)	8.32 (0.10–46.95)	8.21 (0.10–46.95)	8.7 (0.53–29.50)	0.303 ”

‡ *p* value for Student’s t test, * *p* value for chi-squared test, † *p* value for Fisher’s exact test, ” *p* value for Mann–Whitney U test. ICI, Immune Checkpoint Inhibitor; ECOG, Eastern Cooperative Oncology Group; CNS, central nervous system; COPD, Chronic Obstructive Pulmonary Disease; CI, confidence interval; mo, months.

**Table 3 pharmaceuticals-15-00533-t003:** Summary of systemic agents received following immunotherapy discontinuation.

Treatment—*n* (%)	All Patients (*n* = 158)	Nivolumab (*n* = 89)	Atezolizumab (*n* = 69)
No subsequent therapy	95 (60.1)	59 (66.3)	36 (52.2)
Carboplatin combination	8 (5.1)	5 (5.6)	3 (4.3)
Cisplatin combination	2 (1.3)	1 (1.1)	1 (1.4)
Docetaxel	22 (13.9)	9 (10.1)	13 (18.8)
Erlotinib	1 (0.6)	1 (1.1)	0
Gemcitabine	5 (3.2)	4 (4.5)	1 (1.4)
Pemetrexed	2 (1.3)	0	2 (2.9)
Vinorelbine	23 (14.6)	10 (11.2)	13 (18.8)

**Table 4 pharmaceuticals-15-00533-t004:** Best response to treatment.

Response *n* (% [95% CI])	All Patients (*n* = 158)	Nivolumab (*n* = 89)	Atezolizumab (*n* = 69)	*p* Value
ORR	30 (19 [12.8–25.3])	20 (22.5 [13.6–31.3])	10 (14.5 [6.0–23.3])	0.140 *
DCR	69 (43.7 [36.1–51.8])	44 (49.4[38.8–60.0])	25 (36.2 [25.0–40.5])	0.113 *
Complete response	3 (1.9)	3 (3.4)	0	
Partial Response	27 (17.1)	17 (19.1)	10 (14.5)	
Stable disease	39 (24.7)	24 (26.9)	15 (21.7)	
Progressive disease	88 (55.7 [48.2–63.9])	45 (50.6 [40.0–61.1])	43 (62.3 [51.1–75.0])	
Unable to assess	1 (0.6)	0	1 (1.5)	

* *p* value for chi-squared test; ORR, Objective Response Rate; DCR, Disease Control Rate; CI, confidence interval.

**Table 5 pharmaceuticals-15-00533-t005:** Univariate analysis of survival outcomes (PFS and OS).

Parameter	Category	PFS			OS		
		HR	95% CI	*p* Value	HR	95% CI	*p* Value
Age	≤65 vs. >65	1.17	0.83–1.66	0.372	1.04	0.71–1.52	0.835
Sex	Female vs. Male	0.97	0.64–1.49	0.901	0.83	0.52–1.34	0.454
Histology	Squamous vs. non-squamous	1.28	0.90–1.81	0.162	1.48	1.02–2.16	0.040 *
CNS metastasis	No vs. Yes	0.86	0.55–1.36	0.529	1.13	0.67–1.90	0.637
ECOG score	0–1 vs. ≥2	0.48	0.29–0.77	0.003*	0.38	0.22–0.64	<0.001 *
PDL1	− vs. +	1.33	0.84–2.09	0.223	1.09	0.65–1.81	0.745
	Unknown vs. +	0.94	0.63–1.41	0.779	0.95	0.62–1.47	0.829
Smoking status	Never or +10 years former smokers	1.16	0.79–1.71	0.450	1.20	0.79–1.84	0.397
Treatment	Nivo vs. Atezo	0.89	0.62–1.26	0.500	0.92	0.63–1.36	0.685
Line of therapy	2 vs. 3	0.85	0.57–1.28	0.442	0.94	0.60–1.47	0.801
Initial platinum therapy	No vs. Yes	0.49	0.18–1.32	0.157	0.63	0.24–1.81	0.422

* *p* value statistically significant. CNS, central nervous system; ECOG, Eastern Cooperative Oncology Group; PFS, progression-free survival; OS, overall survival; HR, hazard ratio; CI, confidence interval.

**Table 6 pharmaceuticals-15-00533-t006:** Multivariate analysis of survival outcomes (PFS and OS).

Parameter	Category	PFS			OS		
		HR	95% CI	*p* Value	HR	95% CI	*p* Value
Treatment	Nivo vs. Atezo	0.80	0.55–1.17	0.260	0.79	0.52–1.21	0.281
Histology	Squamous vs. non-squamous	1.40	0.96–2.04	0.080	1.68	1.12–2.53	0.012 *
ECOG	0–1 vs. ≥2	0.49	0.30–0.79	0.004 *	0.37	0.22–0.63	<0.001 *

* *p* value statistically significant. CNS, central nervous system; ECOG, Eastern Cooperative Oncology Group; PFS, progression-free survival; OS, overall survival; HR, hazard ratio; CI, confidence interval.

**Table 7 pharmaceuticals-15-00533-t007:** Summary of treatment-related adverse events.

*n*° of Events (% of Patients Affected)	Total		Nivolumab		Atezolizumab		
	Any Grade	Grade 3–5	Any Grade	Grade 3–5	Any Grade	Grade 3–5	*p* Value
Any event	243 (64.55)	23 (14.55)	166 (76.4)	17 (19.10)	77 (49.3)	6 (8.69)	<0.001
Led to temporary drug discontinuation	31 (19.00)	18 (11.39)	24 (25.84)	15 (16.85)	7 (10.14)	3 (4.34)	0.243
Led to definitive drug discontinuation	15 (9.49)	10 (6.32)	10 (11.23)	7 (7.86)	5 (7.24)	3 (4.35)	0.270
Anorexia	21 (13.29)	1 (0.63)	16 (17.98)	0 (0)	5 (7.24)	1 (1.45)	
Arthralgia *	15 (9.49)	0	9 (10.11)	0 (0)	6 (8.69)	0 (0)	
Asthenia/Fatigue	70 (44.30)	2 (1.26)	48 (53.93)	1 (1.12)	22 (31.88)	1 (1.45)	
Diarrhoea *	13 (8.23)	0	8 (9.00)	0 (0)	5 (7.24)	0 (0)	
Hepatotoxicity *	5 (3.16)	2 (1.26)	3 (3.37)	1 (1.12)	2 (2.90)	1(1.45)	
Sickness	14 (8.86)	0	10 (11.23)	0 (0)	4 (5.80)	0 (0)	
Pneumonitis *	10 (6.33)	5 (3.16)	8 (9.00)	4 (4.49)	2 (2.90)	1 (1.45)	
Thyroid dysfunction *	5 (3.16)	0	3 (3.37)	0 (0)	2 (2.90)	0 (0)	
Skin disorder *	43 (27.22)	1 (1.26)	30 (33.70)	1 (1.12)	13 (18.84)	0 (0)	
Pruritus	15 (9.49)	0	10 (11.23)	0 (0)	5 (7.24)	0 (0)	
RASH	20 (12.66)	1 (1.26)	15 (16.85)	1 (1.12)	5 (7.24)	0 (0)	
Vomiting	8 (5.06)	0	4 (4.49)	0 (0)	4 (5.80)	0 (0)	
Other ‡	30 (18.35)	5 (3.16)	19 (21.34)	4 (4.49)	11 (14.49)	1 (1.45)	

* Immune-related adverse events. ‡ Other adverse events: Canker sores, Headache, Conjunctivitis, High creatinine level, Abdominal pain, Oedema, Enterocolitis, Constipation, Myalgia, Mucositis, Neutropenia, Pyrexia, Sensitive polyneuropathy, GERD-related Cough, Thrombopenia.

## Data Availability

The data presented in this study are available on request from the corresponding author. The data are not publicly available, due to privacy restrictions and patient confidentiality.

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
