# Peer review of "Real-World Analysis of Nivolumab and Atezolizumab Efficacy in Previously Treated Patients with Advanced Non-Small Cell Lung Cancer"

_pharmaceuticals, 2022, doi:10.3390/ph15050533_

Round 1

Reviewer 1 Report

In this study, the authors compared the efficacy of two anticancer drugs Nivolumab (anti-PD-1 antibody) and atezolizumab (anti-PD-L1 antibody) for advanced NSCLC patients by using a retrospective cohort study. Their findings show that both drugs exhibit comparable outcomes in overall survival and PFS, but Nivolumab treatment shows a higher rate in adverse events. Generally, this study is properly designed and conducted, the manuscript is well organized, the conclusion is supported by the results, and the analysis is also well performed. These findings provide useful information for understanding the efficacy of and differential pharmacological properties between of the two drugs.

Author Response

Dear reviewer,

we are very grateful for your evaluation.

yours sincerely,

Research team

Reviewer 2 Report

This study aimed to characterize and compare the results (survival, response and safety) between nivolumab and atezolizumab in pretreated (second- and third-line treatment) patients with advanced NSCLC. The results indicated that adverse events occurred in 76.4% of patients in the nivolumab group and 49.3% of patients in the atezolizumab group.

Comments:

  1. Include a better rationalization for choice of nivolumab and ratezolizumab. How nivolumab and ratezolizumab was chosen was not convincing.
  2. The current version of the manuscript lacks a strong discussion. The manuscript should be improved.
  3. The limitation should been more fully developed in the discussion section.

Minor:

Page 3, line 148: All patients (n=58) should be n=158.

Reviewer 3 Report

Real world comparison of nivolumab versus atezolizumab in 3 previously treated patients with advanced non-small cell lung 4 cancer.

Miriam Alonso-García et al. Conducted a monocentric, retrospective study, evaluated the efficacy and safety of nivolumab and atezolizumab in 2nd or 3rd line of NSCLC patients.

The article is well written, data are interesting and well detailed. However, the manuscript is too long and should be shortened in some part. There are several discrepancies regarding number of figure / table.

Title: I guess “comparison of xx versus xx” is not correct, as it is a retrospective analysis, thus there are several major biases in the comparison. Real-world analysyses of nivo and atezo efficacy in…; should be more adequate.

Abstract

Add the number of patients per groups. L22 Adverse events of all grades. Please add "monocentric" in the abstract

Introduction

L52 – median PFS.

L57 – L59 : proportion are unclear: authors should give the % of adenocarcinoma and non-squamous among NSCLC.

L63-L65 : Authors might give some numbers (percentage, differences, ...).

L69-L70 : FDA / EMA : abbreviations

L84-87 : May you please add few details: number of patients, timing of recrutement, … What are differences between your study and these studies? This is important for the reader to understand why your paper is different. 

I guess introduction is well written but too long. Authors should shorten first and second paragraph, and add more details regarding previous studies that evaluated immunotherapy efficacy in real-world conditions.

This is not common to have a Table into the introduction. It should be in the discussion part.

Results

Please move table 2 after the results text.

L127 – L140 – add statistic in  the text and not only numbers (age, CNS M+, histology, …).

L122-L140 : It is possible to shorten this paragraph, which is redundant with the table. Please only focus on main informations.

Table 2 : follow-up in month ? add in the table please. This is the 95%CI into bracket? Add too.

P value = 0,303. Replace with a point.

Efficacy outcomes:

This is not mandatory to repeat the population in the title of the paragraph 2.2. "Efficacy outcomes" is enough.  Same for 2.3

L157-158 : Is it a retrospective evaluation based on the radiological report ? Or imagery had been reviewed by an independent radiologist ? It should be add in the mat & met.

Don’t you have IC95% for DCR and ORR ?

L186 – Figure 2 in the text, Figure 1 in the legend. Same aftern discreptancies regarding number of table / figure.

Figure 1 : Add number of patients / number of events under each curve. Same for all figures.

Discussion

L289 – please add number

L295 – Please add number

Table 1 should be remove in the discussion part.

L307-310 : supplementary results must not be put in the discussion part, but in the results part.

L345 : replace RASH by rash

Materials and methods

Flow chart : excluded patients should be place in the same square. The figure is called figure 1 in the text, but figure 4 in the legend.

Which antibody for PD-L1 status ?

Round 2

Reviewer 2 Report

I think the authors respond the requests properly.

Reviewer 3 Report

I would like to thank the authors for their responses.